A retrospective cohort study of clinical characteristics and outcomes of type 2 diabetic patients with kidney disease

He Xi
Deng Yuanjun
Tian Beichen
Zhao Yixuan
Han Min minhan@tjh.tjmu.edu.cn
Cai Yang caiyang0806@hust.edu.cn caiyang0806@163.com
Department of Nephrology, Tongji Hospital, Tongji Medical College, Huazhong University of Science and Technology , Wuhan , China
Capusa Cristina
Electronic publication date: 2024 Feb 19
Publication date: 2024
Volume: 12
Electronic Location ID: e16915
Received 2023 Oct 2; Accepted 2024 Jan 18
Copyright: ©2024 He et al.
Copyright year: 2024
Copyright holder: He et al.
License: This is an open access article distributed under the terms of the Creative Commons Attribution License, which permits unrestricted use, distribution, reproduction and adaptation in any medium and for any purpose provided that it is properly attributed. For attribution, the original author(s), title, publication source (PeerJ) and either DOI or URL of the article must be cited.
License URL: https://creativecommons.org/licenses/by/4.0/

Keywords: Diabetic nephropathy, Non-diabetic kidney disease, Type 2 diabetes mellitus, Kidney biopsy

Funding: National Natural Science Foundation of China 81770686 81970591 82270725 82000656 82000658 This research was funded by the National Natural Science Foundation of China (No. 81770686, 81970591, 82270725, 82000656, 82000658). The funders had no role in study design, data collection and analysis, decision to publish, or preparation of the manuscript.

==============================
Background

Type 2 diabetes mellitus (T2DM) with chronic kidney disease (CKD) poses a serious health threat and becomes a new challenge. T2DM patients with CKD fall into three categories, diabetic nephropathy (DN), non-diabetic kidney disease (NDKD), and diabetic nephropathy plus non-diabetic kidney disease (DN + NDKD), according to kidney biopsy. The purpose of our study was to compare the clinical characteristics and kidney outcomes of DN, NDKD, and DN + NDKD patients.

Methods

Data on clinical characteristics, pathological findings, and prognosis were collected from June 2016 to July 2022 in patients with previously diagnosed T2DM and confirmed DN and or NDKD by kidney biopsy at Tongji Hospital in Wuhan, China. The endpoint was defined as kidney transplantation, dialysis, or a twofold increase in serum creatinine.

Results

In our 6-year retrospective cohort research, a total of 268 diabetic patients were admitted and categorized into three groups by kidney biopsy. The 268 patients were assigned to DN (n = 74), NDKD (n = 109), and DN + NDKD (n = 85) groups. The most frequent NDKD was membranous nephropathy (MN) (n = 45,41.28%). Hypertensive nephropathy was the most common subtype in the DN+NDKD group (n = 34,40%). A total of 34 patients (12.7%) reached the endpoint. The difference between the Kaplan-Meier survival curves of the DN, NDKD, and DN + NDKD groups was significant (p < 0.05). Multifactorial analysis showed that increased SBP [HR (95% CI): 1.018(1.002–1.035), p = 0.025], lower Hb [HR(95% CI): 0.979(0.961–0.997), p = 0.023], higher glycosylated hemoglobin [HR(95% CI): 1.338(1.080–1.658), p = 0.008] and reduced serum ALB [HR(95% CI): 0.952(0.910–0.996), p = 0.032] were risk factors for outcomes in the T2DM patients with CKD.

Conclusions

This research based on a Chinese cohort demonstrated that the risk of endpoint events differed among DN, NDKD, and DN+NDKD patients. In T2DM patients with CKD, DN patients displayed worse kidney prognosis than those with NDKD or DN + NDKD. Increased SBP, higher glycosylated hemoglobin, lower Hb, and decreased serum ALB may be correlated with adverse kidney outcomes in T2DM patients.

Introduction

More than 500 million people around the world, accounting for over 10.5% of the global adult population are affected by diabetes mellitus (Sun et al., 2022). Type 2 diabetes mellitus (T2DM) comprises the majority of cases. There has been an increase in the population with T2DM from 1990 to 2019 universally in a systematic analysis of T2DM (Ye et al., 2023). The incidence of T2DM among Chinese adults was 12.4%, higher than the world estimate reported by Wang et al. (2021).

Chronic exposure to hyperglycaemia affects the microvasculature in multiple organs, including the kidney, the ocular, the peripheral nervous systems and so on (Barrett et al., 2017). Based on the pathological diagnosis, T2DM patients with chronic kidney disease (CKD) can be classified into diabetic nephropathy (DN), non-diabetic kidney disease (NDKD), and diabetic nephropathy plus non-diabetic kidney disease (DN + NDKD) (Anders et al., 2018). DN affects approximately one-quarter of the diabetic population, which is the primary etiology of end stage renal disease (ESRD) (Faselis et al., 2020). In China, the prevalence of DN was nearly one-fifth of patients with T2DM (21.8%) (Zhang, Kong & Yun, 2020). The prevalence of DN has remained stable while the prevalence of NDKD in T2DM fluctuated greatly. The prevalence of NDKD ranged from 6.5% to 94%, with an average of 41.3% (Zhang et al., 2022). Part of the reason for the difference in prevalence is the discrepancy in clinical practice. This is a reflection of the wide range of considerations by clinicians before a patient undergoes a renal biopsy. NDKD can be either a solitary disease or a coexistence with DN. The diagnosis of NDKD is important since the complete reversal of NDKD is achievable through accurate diagnosis and prompt treatment. The pathological feature and clinical characteristics of T2DM with CKD are likely to change under the conditions of aging population, increasing incidence of infections and malignancies, and the environmental pollution (Prasad et al., 2023). Our understanding of the pathophysiologic mechanisms of T2DM with CKD has progressed as we continue to refine our classification of the pathologic types of T2DM with CKD. The commonly reported variables of NDKD, including DM shorter duration, lower glycosylated hemoglobin, absence of retinopathy, lower blood pressure, hematuria, higher proteinuria, higher hemoglobin, and lower serum creatinine were considered as the risk factors for kidney function progression in previous studies (Horvatic et al., 2014; Jing et al., 2021; Prasad et al., 2023). To our knowledge, the etiology and demographic data is limited in South China, and there have been few studies comparing the prognosis of T2DM patients with CKD based on the classification of DN, NDKD, and DN + NDKD.

Therefore, it is imperative to reassess CKD in T2DM and know the spectrum of T2DM with CKD considering the huge burden of T2DM and diabetes-related kidney diseases in China. Our study used the cohort in our center to further evaluate the differences in prognosis among DN, NDKD and DN + NDKD patients. The endpoint was defined as kidney transplantation, dialysis or a twofold increase in serum creatinine. Thus, it will be possible to clarify whether patients with DN have a worse prognosis than patients with NDKD and to investigate prognostic risk factors. By managing the associated risk factors, our research is expected to provide preventive or therapeutic interventions for T2DM patients with CKD. Effective prevention can reduce the disease burden in patients with CKD and improve their quality of life and prognosis.

Materials & Methods

Study design and patients

Patients with previously diagnosed T2DM with CKD by kidney biopsy were enrolled from 1 June 2016 to 31 July 2022, at Tongji Hospital, Wuhan, China in this retrospective study. The inclusion criteria for this study were as followed: (i) age >18 years; (ii) clinical diagnosis of T2DM; (iii) underwent kidney biopsy. The exclusion criteria included the items below: (i) ESRD diagnosed before kidney biopsy or estimated glomerular filtration rate (eGFR) <15 ml/min per 1.73 m2 (exclusion: seven); (ii) patients with other types of diabetes mellitus or combined malignancy (exclusion: 18); (iii) severe clinical data deficit (exclusion: 22); (iii) kidney transplantation, acute kidney injury and urinary tract infection (exclusion: 0). The endpoint was defined as kidney transplantation, dialysis or a twofold increase in serum creatinine. This study adhered to the tenets of the Declaration of Helsinki declaration. Informed consent was waived by the Ethics Review Board of Tongji Hospital, Tongji Medical College, Huazhong University of Science and Technology (No. TJ-IRB20210929).

Data acquisition

From the electronic medical record, we extracted demographic data (age and sex), blood pressure values (systolic blood pressure (SBP) and diastolic blood pressure (DBP)), and medication history, with all original examination dates obtained from patients’ initial admissions. All original examination dates were derived from patients initial admission. Clinical data included hemoglobin (Hb), serum albumin (ALB), 24 h proteinuria, eGFR, serum creatinine (Scr), hemoglobin A1C (HbA1C), immunoglobulin G (IgG), immunoglobulin A (IgA), immunoglobulin M (IgM), complement 3 (C3), and complement 4 (C4).

Pathological examination

The kidney puncture tissues were examined by light microscopy, immunofluorescence and electron microscopy. The pathological diagnosis of DN was based on the 2010 version of the pathologic classification of diabetic nephropathy (Tervaert et al., 2010). A diagnosis of DN is confirmed by one of the following conditions: class I, glomerular basement membrane thickening; class II, mild (IIa) or severe (IIb) mesangial expansion; Class III, nodular sclerosis (Kimmelstiel-Wilson lesions): at least one glomerulus with nodular increase in mesangial matrix (Kimmelstiel-Wilson); Class IV, more than 50% global glomerulosclerosis. Light microscopy, immunofluorescence, and electron microscopy were used to diagnose NDKD based on characteristic changes. Two experienced independent pathologists reviewed all biopsy specimens.

Statistical analysis

SPSS (version 25.0, IBM, US) software and R software (version 4.2.2, R Core Team, 2023) were used to analyze the full analysis set. Data were presented as median (interquartile range, IQR) or mean (standard deviation, SD) after normality tests on continuous variables, and as numbers and percentages on categorical variables. Missing values were imputed by predictive mean of closest points. The one-way ANOVA test, the Kruskal-Wallis test, and the Ç2 test were used to assessed differences. Endpoints were defined as dialysis, death, or twofold increase in serum creatinine. Kaplan–Meier analysis and Cox regression analysis were utilized to perform time-to-event analysis. Kaplan–Meier survival curves were plotted for patients in the DN, NDKD, and DN + NDKD groups. The results were compared using log-rank tests. Relevant risk factors and covariates with p values < 0.1 were included in Cox regression proportional risk models. Covariates included indicators that met the requirements or were clinically significant after univariate analysis. Variables were entered into the Cox model through backward entry method. The validity was determined by testing the chi-square value of the Cox model.

Results

Baseline characteristics of three groups

From a cohort of 315 T2DM patients underwent kidney biopsy between 2016 and 2022, we excluded 47 non-compliant patients after applying exclusion criteria. The remaining 268 patients included 74 in the DN group, 109 in the NDKD group, and 85 in the DN + NDKD group, or 27.61%, 40.67%, and 31.72% of the cohort, respectively (Fig. 1).

Figure 1 Flow chart of participant selection in this study.

From the 315 patients in our single center, we screened patients who met the inclusion criteria. A total of 268 participants were included and divided into three groups based on kidney puncture results.

Table 1 presents the baseline characteristics of our study. 180 (67.2%) of the 268 patients with inclusion criteria were male and 88 (32.84%) were female. The median (±IQR) age of all those included in the criteria was 52.50 ±15 years, varying from 26 to 73 years. The median age of the DN group was 50 years and it was 54 years (IQR 15) in the NDKD group, and 53 years (IQR 12) in the DN + NDKD group. We followed patients for an average of 16.44 months. The mean duration of DM was 60 months, in order of NDKD, DN + NDKD, DN, from shortest to longest.

Table 1 Baseline characteristics and drug treatment of patients in the DN group, NDKD group and DN + NDKD group.

Characteristic	Overall (n = 268)	DN (n = 74)	NDKD (n = 109)	DN + NDKD (n = 85)	p-value	
Age (median [IQR])	52.50 (15.00)	50.00 (18.00)	54.00 (15.00)	53.00 (12.00)	0.133	
Gender = male (%)	180.00 (67.20)	55.00 (74.30)	59.00 (54.10)	66.00 (77.60)	0.001**	
Cigarette (%)	60.00 (22.40)	21.00 (28.40)	18.00 (16.50)	21.00 (24.70)	0.138	
family history of diabetes (%)	23.00 (8.58)	12 (16.22)	7 (6.42)	4 (4.71)	0.021*	
RASi (%)	184 (68.70)	53.00 (71.60)	80.00 (73.40)	51.00 (60.00)	0.101	
Immunosuppressant (%)	23.00 (8.60)	3.00 (4.10)	16.00 (14.70)	4.00 (4.70)	0.013*	
Glucocorticoid (%)	51 (19.00)	1.00 (1.40)	40.00 (36.70)	10.00 (11.80)	0.000**	
Insulin (%)	122 (45.50)	46.00 (62.20)	37.00 (33.90)	39.00 (45.90)	0.000**	
Follow-up time (mean (SD))	16.44 (13.74)	15.32 (10.00)a	16.44 (19.00)b	15.00 (11.00)a	0.009**	
The duration of DM (median [IQR])	36.00 (82.25)	60.00 (111.00)a	24.00 (57.00)b	42.00 (79.50)a	0.007**	
DR (%)	43.00 (16.00)	27.00 (36.50)	4.00 (3.70)	12.00 (14.10)	0.000**	
Sbp (mean (SD))	138.43 (22.81)	141.70 (22.74)	136.97 (21.30)	137.46 (24.66)	0.348	
Dbp (mean (SD))	86.93 (13.68)	87.99 (12.73)	86.97 (13.76)	85.94 (14.46)	0.643	
Hb (mean (SD))	125.82 (23.73)	122.01 (5.75)a	129.69 (20.61)b	125.04 (25.05)	0.048*	
RBC (median [IQR])	4.29 (1.00)	4.16 (1.13)a	4.34 (1.01)b	4.30 (1.13)	0.043*	
24 h proteinuria (median [IQR])	2484.76 (4431.73)	3438.53 (5331.20)	2148.00 (4583.10)	1925.80 (3860.20)	0.055	
24 h urine protein >3.5 g (%)	87.00 (32.46)	30.00 (40.54)	34.00 (31.19)	23.00 (27.06)	0.126	
Urinary sediment RBC (median[IQR])	30.15 (49.84)	25.15 (28.15)a	41.30 (87.95) b	14.20 (63.62)a	0.004**	
BUN (median[IQR])	7.29 (3.97)	8.21 (4.53)a	6.60 (3.00) b	7.97 (4.08)a	0.000**	
Scr (median [IQR])	103.50 (77.75)	115.00 (84.38)a	89.00 (48.50)b	122.00 (100.50)a	0.000**	
EGFR (median[IQR])	64.95 (48.35)	59.10 (50.48)a	78.00 (41.95)b	55.50 (44.25)a	0.000**	
ALB (median[IQR])	37.80 (13.48)	34.95 (11.55)	38.80 (15.65)	39.70 (10.90)	0.139	
Blood glucose (median [IQR])	8.32 (4.47)	8.98 (5.49)	8.26 (3.80)	8.23 (4.41)	0.162	
Glycosylated hemoglobin (median [IQR])	6.85 (1.50)	7.60 (1.80)a	6.60 (1.10)b	6.80 (1.00)b	0.002**	
TC (median [IQR])	4.77 (1.99)	4.90 (1.74)	4.80 (2.20)	4.58 (1.80)	0.233	
TG (median [IQR])	2.51 (2.48)	2.72 (3.01)	2.61 (2.47)	2.39 (2.23)	0.265	
HDL-C (median[IQR])	0.98 (0.38)	0.96 (0.34)	1.04 (0.38)a	0.93 (0.39)b	0.043*	
LDL-C (median[IQR])	2.65 (1.27)	2.60 (1.10)	2.74 (1.32)	2.45 (1.42)	0.448	
IgG (mean (SD))	9.87 (3.74)	9.70 (4.40)	9.60 (5.55)	10.49 (3.94)	0.059	
IgA (median [IQR])	2.54 (1.43)	2.45 (1.49)	2.64 (1.61)	2.53 (1.19)	0.730	
IgM (median[IQR])	0.97 (0.60)	0.98 (0.61)	1.05 (0.68)	0.89 (0.37)	0.101	
C3 (median [IQR])	0.96 (0.25)	0.92 (0.23)a	1.03 (0.26)b	0.94 (0.18)a	0.003**	
C4 (median [IQR])	0.26 (0.09)	0.27 (0.09)	0.26 (0.10)	0.25 (0.09)	0.609	
Notes.

Data are presented as medians with ranges, or counts and percentages. a and b represent instances where there are significant differences between a and b.

* P < 0.05.

** P < 0.01.

*** P < 0.001.

Abbreviations DR diabetic retinopathy

DM diabetes mellitus

UACR urinary albumin/creatinine ratio

Sbp systolic blood pressure

Dbp diastolic blood pressure; Hb

BUN Blood Urea Nitrogen

Scr serum creatinine

EGFR estimated glomerular filtration rate

ALB albumin

TC total cholesterol

TG triglyceride

HDL-C high density lipid-cholesterol

LDL-C low density lipid-cholesterol

Immunosuppressants and glucocorticoids were most commonly used in the NDKD group. Insulin was the predominant treatment in the DN group. Statistically meaningful differences were observed in the three groups with regard to gender (p = 0.001), family history of diabetes (p = 0.021), duration of T2DM (p = 0.007), diabetic retinopathy (p < 0.001), red blood cell count (p = 0.043), urinary sediment red blood cell count (p = 0.004), glycosylated hemoglobin (p = 0.002), HDL (p = 0.043), and C3 (p = 0.003) (Table 1).

Pathological characteristics of kidney alterations in T2DM patients

Typical DN pathologic images are shown (Fig. 2). The most prevalent pathological type in the NDKD group was membranous nephropathy (n = 45). Other subtypes within the NDKD category were IgA nephropathy (n = 26), hypertensive nephropathy (n = 10), Henoch-Schoenlein purpura nephritis (n = 5), obesity-related glomerulopathy (n = 4), focal segmental glomerulosclerosis (n = 3), light chain deposition disease (LCDD) (n = 1), kidney amyloidosis (n = 1), tubulointerstitial nephritis (n = 2), thrombotic microangiopathy (n = 2), hepatitis B virus-related nephropathy (n = 1), sclerosing glomerulonephritis (n = 3), minimal change disease (n = 2), proliferative glomerular lesions (n = 4). Hypertensive nephropathy (n = 34) was the dominant subtype, followed by IgA nephropathy (n = 15) in the DN + NDKD group (Table 2, Fig. S1).

The compassion of the cumulative incidence of endpoints in T2DM with CKD patients

Our average follow-up in this cohort was 16.44 months. The study’s endpoints were all-cause death, kidney transplantation, dialysis, and a twofold increase in serum creatinine. For an overall endpoint frequency of 12.7%, a total of 34 patients met the endpoint. After analyzing the incidence of the endpoints, our study found that the number of patients with endpoints were 13 in the DN group, nine in the NDKD group and 12 in DN + NDKD group, with proportions of 17.57%, 8.26%, 14.12%. Endpoint incidence was notably greater in the DN group compared to the other groups (p < 0.05) (Fig. 3). The median survival time remained at 52.0 months for NDKD and 34.5 months for DN. The median survival time of DN + NDKD group can’t be estimated as there were few endpoints in this group and most survival times correspond to survival probabilities greater than 0.5. One-year survival rate of kidney in each group were 88.8%, 97.4%, 87.7% in the DN, NDKD and DN + NDKD group.

Figure 2 Pathologic manifestations of DN.

Table 2 Comparison of pathological characteristics between NDKD and DN + NDKD groups.

Pathological characteristic	NDKD (109)	DN + NDKD (85)	
IgA nephropathy	26	15	
Membranous nephropathy	45	12	
Hypertensive nephropathy	10	34	
Henoch-Schoenlein purpura nephritis	5	0	
Obesity-related nephropathy	4	0	
Focal segmental glomerulosclerosis	3	3	
Light chain deposition disease	1	0	
Kidney amyloidosis	1	0	
Tubulointerstitial nephritis	2	8	
Thrombotic Microangiopathy	2	1	
Hepatitis B virus-related nephropathy	1	1	
Sclerosing glomerulonephritis	3	2	
Minimal change disease	2	1	
Proliferative glomerular lesions	4	1	
Acute tubular necrosis	0	2	
HCV associated glomerulonephritis	0	1	
Post-infectious glomerulonephritis	0	1	
Crescentic glomerulonephritis	0	3	

Figure 3 Comparison of renal survival rate in the DN group, NDKD group and DN + NDKD group.

Prognostic factors for endpoints

The proportional hazards (PH) assumption tests were conducted for the variables in the endpoints. The test results indicated that all variables satisfied the PH assumption. A multivariate Cox proportional hazards regression model included baseline variables that were deemed clinically relevant or univariately associated with the outcomes. The final model was simplified by careful selection of variables based on the number of events available (Table 3). Lower serum ALB [HR (95% CI): 0.685 (0.559–0.839), p < 0.001], 24 h proteinuria [HR (95% CI): 0.999 (0.999–1.000), p = 0.006], and increased SBP [HR (95% CI): 1.047 (1.006–1.089), p = 0.024] and age [HR (95% CI): 0.890 (0.802–0.988), p = 0.028] were determined to be important contributors to adverse kidney outcomes in the DN group by multivariate Cox regression analysis (Table 4). NDKD patients with higher 24 h proteinuria [HR (95% CI): 1.000 (1.000–1.001), p = 0.019] and decreased C3 [HR (95% CI): 0.001 (0.000–0.356), p = 0.021], were at increased risk for adverse kidney effects (Table 4). Multivariate Cox regression results showed that serum ALB [HR (95% CI): 0.828 (0.724–0.947), p = 0.006], Scr [HR (95% CI): 1.011 (1.005–1.018), p = 0.001], IgM [HR (95% CI): 13.708 (3.611–52.034), p < 0.001], SBP [HR (95% CI): 1.050 (1.007–1.094), p = 0.021], age [HR (95% CI): 0.851 (0.756–0.958), p = 0.008] were significant risk indicators for the endpoint event in the cohort of the DN + NDKD group (Table 4). T2DM patients with CKD showed that SBP [HR (95% CI): 1.018 (1.002–1.035), p = 0.025], Hb [HR (95% CI): 0.979 (0.961–0.997), p = 0.023], ALB [HR (95% CI): 0.952 (0.910–0.996), P = 0.032], glycosylated hemoglobin [HR (95% CI): 1.338 (1.080–1.658), p = 0.008], were independent indicators of risk for the adverse kidney outcomes.

Table 3 Univariate Cox regression analyses for endpoints.

(A) The univariate Cox analysis results of the DN and NDKD group.	
Characteristic	DN	NDKD	
	HR (95% CI)	p-value	HR (95% CI)	p-value	
Age	1.008 (0.957–1.062)	0.759	1.026 (0.958–1.100)	0.461	
Gender (male)	1.052 (0.283–3.904)	0.940	0.686 (0.166–2.830)	0.602	
SBP	1.047 (1.019–1.076)	0.001**	0.99 (0.957–1.025)	0.580	
Hb	0.935 (0.899–0.973)	0.001**	0.976 (0.939–1.015)	0.221	
Urinary sediment RBC	1.027 (1.008–1.048)	0.007**	0.993 (0.975–1.011)	0.452	
Serum Alb	0.792 (0.704–0.893)	0.000***	0.93 (0.859–1.006)	0.071	
Scr	1.013 (1.005–1.022)	0.002**	0.997 (0.98–1.015)	0.777	
24 h urine protein	1.00 (1.00–1.00)	0.047*	1.00 (1.00–1.00)	0.050*	
TC	1.093 (0.721–1.657)	0.676	1.004 (0.652–1.545)	0.986	
TG	0.813 (0.65–1.017)	0.07	1.01 (0.818–1.246)	0.928	
HDL	1.606 (0.44–5.866)	0.473	2.389 (0.325–17.535)	0.392	
LDL	1.163 (0.642–2.107)	0.618	0.761 (0.382–1.513)	0.436	
IgG	0.757 (0.588–0.974)	0.03*	0.759 (0.581–0.991)	0.043*	
IgA	1.428 (0.818–2.492)	0.21	0.586 (0.263–1.306)	0.191	
IgM	0.523 (0.16–1.709)	0.283	1.568 (0.33–7.462)	0.572	
C3	0.173 (0.005–6.185)	0.336	0.020 (0.000–0.937)	0.046*	
(B) The univariate Cox analysis results for the DN+NDKD group and whole cohort.	
Characteristic	DN + NDKD	All patients	
	HR (95% CI)	p-value	HR (95% CI)	p-value	
Age	1.010 (0.942–1.083)	0.783	1.006 (0.973–1.041)	0.725	
Gender (male)	0.398 (0.105–1.516)	0.177	0.845 (0.408–1.749)	0.65	
SBP	1.032 (1.003–1.062)	0.028*	1.024 (1.009–1.04)	0.002**	
Hb	0.976 (0.953–0.999)	0.042*	0.963 (0.947–0.979)	0.000***	
Urinary sediment RBC	1.00 (1.00–1.00)	0.769	1.00 (1.00–1.00)	0.728	
Serum Alb	0.931 (0.872–0.993)	0.029*	0.93 (0.895–0.967)	0.000***	
Scr	1.004 (1.002–1.007)	0.003**	1.003 (1.001–1.005)	0.001**	
24 h urine protein	1.00 (1.00–1.00)	0.463	1.00 (1.00–1.00)	0.024*	
TC	1.056 (0.729–1.528)	0.774	0.952 (0.765–1.185)	0.662	
TG	0.947 (0.728–1.232)	0.684	0.933 (0.818–1.065)	0.304	
HDL	1.734 (0.277–10.838)	0.556	1.377 (0.511–3.708)	0.527	
LDL	1.214 (0.771–1.912)	0.402	0.966 (0.699–1.334)	0.833	
IgG	0.949 (0.789–1.141)	0.579	0.897 (0.804–1.002)	0.054	
IgA	1.07 (0.583–1.964)	0.828	0.876 (0.64–1.199)	0.41	
IgM	4.049 (1.537–10.668)	0.005**	1.992 (1.119–3.549)	0.019*	
C3	0.142 (0.002–11.771)	0.386	0.062 (0.008–0.49)	0.008**	
Notes.

* p < 0.05.

** P < 0.01.

*** P < 0.001.

Table 4 Multivariate Cox regression analyses for endpoints.

(A) DN.	
Characteristic	HR (95% CI)	p-value	The Chi-square values	p-value	
ALB	0.685 (0.559–0.839)	<0.001***	36.846	<0.001***	
24 h proteinuria	0.999 (0.999–1.000)	0.006*	
SBP	1.047 (1.006–1.089)	0.024*	
age	0.890 (0.802–0.988)	0.028*	
(B) NDKD.	
Characteristic	HR (95% CI)	p-value	The Chi-square values	p-value	
Urinary sediment RBC	0.975 (0.948–1.003)	0.082	9.896	0.042*	
24 h proteinuria	1.000 (1.000–1.001)	0.019*	
IgG	0.803 (0.584–1.105)	0.178	
C3	0.001 (0.000–0.356)	0.021*	
(C) DN + NDKD.	
Characteristic	HR (95% CI)	p-value	The Chi-square values	p-value	
ALB	0.828 (0.724–0.947)	0.006**	53.626	<0.001***	
Scr	1.011 (1.005–1.018)	0.001**	
IgM	13.708 (3.611–52.034)	<0.001***	
SBP	1.050 (1.007–1.094)	0.021*	
age	0.851 (0.756–0.958)	0.008**	
(D) Total patients.	
Characteristic	HR (95% CI)	p-value	The Chi-square values	p-value	
SBP	1.018 (1.002–1.035)	0.025*	50.029	<0.001***	
Hb	0.979 (0.961–0.997)	0.023*	
C3	0.133 (0.014–1.228)	0.075	
ALB	0.952 (0.910–0.996)	0.032*	
Glycosylated hemoglobin	1.338 (1.080–1.658)	0.008**	
Notes.

* p < 0.05.

** P < 0.01.

*** P < 0.001.

Discussion

T2DM with CKD patients were divided into three groups in this study according to kidney biopsy. Our results found that 40.67% of biopsied T2DM patients were diagnosed with NDKD and the incidence of DN + NDKD was more than one-third (31.72%) of T2DM patients. Previous study showed that the prevalence of NDKD averaged 41.3% (Zhang et al., 2022) and prevalence in the DN + NDKD group varied from 4.7% to 19.72% (Fontana et al., 2021; Liu et al., 2016; Shadab et al., 2022). The above study demonstrates that a high proportion of T2DM patients with CKD still have NDKD, and that there is a great heterogeneity in the prevalence.

MN was the most prevalent with 41.28%, followed by IgA nephropathy with 23.85% in our study, consistent with the findings reported by Wang et al. (2019). But some researchers conclude that the major pathologic subtype of NDKD is IgA nephropathy (Byun et al., 2013; Zhou et al., 2008). Regional and ethnic differences, as well as the mechanism of kidney pathologic diagnosis, may contribute to the pathologic distribution of the NDKD group.

Progression of T2DM, poor glycemic control, DR, deterioration of kidney function, hematuria, hypertension can guide to differentiation between DN and NDKD in many studies (Li et al., 2020; Popa et al., 2021; Saini, Kochar & Poonia, 2021), which were consistent with our findings. Pallayova et al. (2015) found that a strong predictor of NDKD was low serum HbA1c level. The ratio of glycated albumin to HbA1c, according to Wang et al. (2017), was better biopsy-proven DN indicators than HbA1c. DN and DR, as the two most important microvascular diseases of T2DM, share many pathophysiologic and pathologic similarities. DR was closely correlated with DN (±NDKD), and the absence of DR was a highly predictive of NDKD (Lin et al., 2018), while Kritmetapak et al. (2018) found that in a multivariate analysis DR was not an independent predictor (Kritmetapak et al., 2018) and the association between DN and DR is not exactly parallel conducted by Li et al. (2021). Usually lack of DR is predictive of NDKD, but does not exclude DN.

The hemoglobin levels in the DN patients were markedly lower as opposed to the NDKD patients. In the primal stages of kidney disease, studies have revealed that CKD patients with T2DM may become anemic (Xie et al., 2023). A recent cohort study in Japan showed that serum Hb concentration, reflecting the onset of kidney fibrosis, may be useful in predicting the development of DN (Yamanouchi et al., 2022). Ito et al. (2021) considered that because of severe interstitial fibrosis and tubular atrophy, DN is associated with anemia and anemia may aid in clinical differentiation between isolated DN and NDKD. Furthermore, erythrocytes deformability and lifespan are also reduced by chronic inflammation and advanced glycation end products (Tsai & Tarng, 2019).

In our study, HDL levels differed at baseline levels, but did not affect the prognosis. Nevertheless low HDL-C and high TG levels, in an Italian study, were considered independent risk factors for DN prognosis over 4-year period (Russo et al., 2016). The cause of high TG and low HDL-C may be caused by metabolic syndrome, and may result from underlying insulin resistance. Multiple aspects of kidney function, including kidney hemodynamics and tubular function, are adversely affected by insulin resistance (Artunc et al., 2016).

The pathological classification of CKD with T2DM, in our results, was significantly associated with kidney prognosis. Sun et al. (2023) have also shown that DN patients had relatively poorer outcomes than NDKD. DN patients have a faster progression to ESRD than other CKD etiologies, requiring earlier kidney replacement therapy, which results in a significant health and economic burden.

To further investigate potential predictors of the kidney endpoint of T2DM patients, we conducted the multivariate Cox regression analyses in T2DM patients with CKD. We found that lower serum ALB, elevated SBP, glycosylated hemoglobin and Hb were independent risk factors for the endpoint of all patients. In addition, we explored factors affecting kidney prognosis through subgroup analyses, including those mentioned above. ALB, 24 h proteinuria, SBP and age were the most powerful risk factors for adverse kidney outcomes of the DN group in our analysis. These factors largely correspond to the risk factors traditionally linked to DN. Hypoalbuminemia may reflect multiple diseases: cirrhosis, malnutrition, kidney diseases and chronic inflammation (Aldebeyan et al., 2017; Efremova et al., 2023; Sheinenzon et al., 2021). Therefore, hypoalbuminemia may influence the progression of CKD through the mechanisms described above. In Japanese patients with CKD, there was a negative and non-linear relationship between ALB and the decline in kidney prognosis (Cheng et al., 2023). Moreover, hypertension, identified as an independent predictor of microvascular complications (Asghar et al., 2023), induces oxidative stress and inflammation in the kidney (Lopes de Faria, Silva & Lopes de Faria, 2011). With the exception of age, most of these risk factors are controllable, which is particularly critical for the management of DN. It is known that typical lesions of diabetic nephropathy include glomerular hyperfiltration and podocyte injury. Among the mechanisms of podocyte injury are lipotoxicity, oxidative stress, mitochondrial damage, and autophagy (Li et al., 2023b; Nagata, 2016). In fact, the molecular mechanism of DN is complex and many pathways are involved in DN development and progression in a hyperglycemic environment including polyol, hexosamine, PKC, and AGE pathways. These indicators may be involved in the progression of diabetic nephropathy through the mechanisms described above.

Surprisingly, C3 was identified as putative risk features for the endpoint in the NDKD group. It is universally acknowledged that C3 is an important part of the complement system and has three distinct modes of activation: classic, lectin, and alternative. Li et al. (2023a) found that MN patients with 24 h proteinuria over 0.75 g or serum albumin below 35g/l had persistent low serum C3. Previous studies have also suggested that low serum C3 predicted poor kidney outcomes (Tsai, Wu & Chen, 2019). Besides, Rajasekaran et al. (2023) noted that complement markers in kidney biopsies of IgAN patients were related to disease activity and predicted poor kidney prognosis. With regard to the composition of the pathology types in the NDKD and DN + NDKD group, we supposed that the mechanisms of MN and IgAN have complement involvement, which influences the prognosis of the NDKD group and differences in the composition of pathologic types between the two groups also led to different prognostic factors in the NDKD and DN + NDKD groups. That, to some extent, could give some explanation of why lower C3 predicts ESKD in NDKD group. It is worth mentioning that DN group have lower levels of C3 compared to the NDKD and DN + NDKD group, but it was not statistically significant in univariate and multivariate Cox analyses, which may be composed of multiple reasons. On the one hand, it has been suggested that the effects of C3 on kidney outcomes may be counteracted by other factors such as blood lipids (Zhang et al., 2018). On the other hand, C3c, but not C3, may be associated with worse kidney prognosis (Li et al., 2022). In conclusion, there may be confounding or mediating variables between C3 and the kidney prognosis of DN patients. More experimental and clinical evidence is needed to validate the relationship between C3 and kidney prognosis of DN patients.

Additionally, ALB, Scr, IgM, SBP and age were possible risk elements for the outcome in patients with DN + NDKD. Beside the traditional risk factors, we noted the appearance of IgM. Prior research had found that IgM may cause damage through activation of the glomerular thylakoid complement cascade mediated by the classical immune complex (Mubarak & Kazi, 2012). Al Romaili et al. (2019) found that IgM deposition in minimal change disease (MCD) showed statistical association with CKD and IgM may play a role in MCD. While there was only one case of this type in the DN + NDKD group we studied. Further studies are needed to verify the causal relationship between elevated IgM levels and kidney prognosis. Taken together, these indicators were associated with declining kidney function.

Predictive models for diabetes-related kidney disease have been developed by many researchers. But most of the models are not applicable to the Chinese population due to patient populations, and study methodology. Riphagen et al. (2015) chose two clinical end points: development of (micro)albuminuria and progressive kidney function loss. The inclusion population of Anderson et al. (2021) included some patients without diabetes mellitus. Meanwhile, Chen, Chen & Jiang (2022) focused on analyzing the risk factors for three different endpoint events by constructing Cox regression models. Strengths of the present study are that it focused on subgroup analysis of T2DM populations, explored different prognostic factors for DN, NDKD and DN + NDKD, and established three group different prediction models for Chinese populations.

There are some limitations of this study and the analysis of the results may be biased. We analyzed risk factors affecting prognosis using only a single-center cohort of individuals from China. Because the epidemiology of T2DM patients with CKD shows significant global variation, it may affect the generality of the application, but it may be useful to physicians in the region in their daily practice. We hope to follow up with a multi-center, large sample size study. Next, in our cohort, there was insufficient follow-up time for some patients, but it is emphasized that the majority of patients enrolled in our study were not newly diagnosed with diabetes at the start of the follow-up period. This aspect partially mitigated the limitations of our relatively short follow-up duration. Additionally, one of the inevitable problems with clinical retrospective studies is the presence of bias: exclusion of patients due to excessive missing information may create a selection bias. In recent years, the use of new drugs has greatly improved the prognosis of DN, thus further comparisons of the prognosis of the three groups after treatment are needed.

Conclusions

In conclusion, this respective single-center cohort research based on a Chinese population demonstrated that the risk of endpoint events differed among DN, NDKD, and DN + NDKD groups. Patients with DN presented worse kidney prognosis than those with NDKD or DN + NDKD. In the T2DM patients with CKD, it has been found that, low serum ALB, Hb, higher glycosylated hemoglobin and increased SBP, were independent risk parameters for the occurrence of endpoint events. Therefore, it is crucial to focus on the DN group and implement early preventive or therapeutic measures in order to delay the occurrence of these endpoints.

Supplemental Information

Figure S1 Representative figures of NDKD pathology

(A–C) IgA nephropathy (IgAN): light microscopy reveals mesangial matrix expansion or mesangial hypercellularity (yellow arrow). Immunofluorescence reveals the deposition of IgA in the glomerular mesangial area. Electron microscopy reveals the deposition of electron-dense material in the mesangial region (red arrow). (D–F) Membranous nephropathy (MN): light microscopy shows glomerular basement membrane thickening (yellow arrow). Immunofluorescence reveals coarse granular staining of IgG along the glomerular capillary loop and electron microscopy shows subepithelial electron-dense deposits and diffused fusion of the foot processes of podocytes (red arrow). (G–I) Kidney amyloidosis: light microscopy shows extensive deposits of pink amorphous eosinophilic material (yellow arrow). Immunofluorescence reveals deposits of λ along the mesangial glomeruli and capillary walls. Electron microscopy reveals randomly arrayed nonbranching fibrils (8–10 nm in diameter). (J–L) Sclerosing glomerulonephritis: more than 50% of the glomeruli show global sclerosis (yellow arrow). Furthermore, immunofluorescence shows non-specific C3 deposition in sclerotic areas and electron microscopy shows no electron-dense material deposition. (M–O) Crescentic glomerulonephritis: light microscopy reveals cellular crescent formation (yellow arrow). In anti-GBM positive crescentic nephritis, immunofluorescence shows linear deposition of IgG along capillary walls and electron microscopy showed no electron-dense material deposition. (P) Obesity-related nephropathy: light microscopy shows glomerulomegaly (yellow arrow). (Q–R) Hypertensive nephropathy: light microscopy reveals hyaline degeneration of glomerular arterioles and ischemic glomerular sclerosis (yellow arrow). Electron microscopy shows no electron-dense material deposition and partial fusion of the foot processes of podocytes. (S) Tubulointerstitial nephritis: light microscopy shows tubulitis and renal interstitial edema with inflammatory cell infiltration, accompanied by local distribution of kidney tubular epithelial cell injury and fibrosis. (T–U) Focal segmental glomerulosclerosis (FSGS): light microscopy reveals lesions of focal and segmental glomerular sclerosis (yellow arrow). Electron microscopy shows no electron-dense material deposition and diffuse fusion of the foot processes of podocytes.

Supplemental Information 2 Code for making survival analysis chart

Data S1 Raw data

All patients who enrolled in the inclusion criteria. These data were used for statistical analysis to compare the clinical characteristics and kidney outcomes.

We thank all the investigators and the study participants for their invaluable work.

Additional Information and Declarations

Competing Interests

Author Contributions

Human Ethics

Data Availability

The authors declare there are no competing interests.

Xi He performed the experiments, analyzed the data, prepared figures and/or tables, and approved the final draft.

Yuanjun Deng analyzed the data, authored or reviewed drafts of the article, and approved the final draft.

Beichen Tian performed the experiments, prepared figures and/or tables, and approved the final draft.

Yixuan Zhao performed the experiments, prepared figures and/or tables, and approved the final draft.

Min Han conceived and designed the experiments, authored or reviewed drafts of the article, and approved the final draft.

Yang Cai conceived and designed the experiments, authored or reviewed drafts of the article, and approved the final draft.

The following information was supplied relating to ethical approvals (i.e., approving body and any reference numbers):

The Ethics Review Board of Tongji Hospital, Tongji Medical College, Huazhong University of Science and Technology approval to carry out the study (No. TJ-IRB20210929).

The following information was supplied regarding data availability:

The raw data are available in the Supplementary Files.

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
