# Peer review of "A retrospective cohort study of clinical characteristics and outcomes of type 2 diabetic patients with kidney disease"

_PeerJ, doi:10.7717/peerj.16915_

## Round 0.1 · original submission · Major Revisions

Limitations in the study methodology exist and authors are kindly asked to resolve it.

Reviewer 1 ·

Basic reporting

1. The use of term ‘kidney’ rather than ‘renal’ is recommended to be used in most of the circumstances. In this regard the following article should be consulted for more guidance https://www.kidney-international.org/article/S0085-2538%2820%2930233-7/fulltext
2. The term ‘Diabetic kidney disease’ (DKD) is defined by the coexistence of CKD with diabetes mellitus, regardless of the histologic diagnosis (if it exists). Therefore, considering that in the current study the histologic diagnoses are known, diabetic nephropathy is probably more suitable to be used instead of DKD.
3. Hepatitis B/C virus-related nephropathy or obesity related nephropathy are not histologic diagnoses, while ANCA-associated GN is a crescentic GN but is separately presented from Crescentic GN. I recommend the use of either the histologic pattern of injury, or the clinico-pathologic diagnosis.
4. The manuscript needs linguistic revisions in regard with the used English language. Some examples where the language could be improved include lines 63, 70, 87-89, 186, 221.

Experimental design

1. Could you please explain why you excluded subjects with urinary tract infection (even acute cystitis or asymptomatic bacteriuria?) and malignancies – these exclusions may have influenced the final number of subjects from each group. It may also be useful if you write the exact number of subjects you excluded for each cause.
2. Based on which characteristics did the authors differentiate subjects with DKD + superimposed Hypertensive nephropathy from those with only DKD?
3. How do authors explain the higher C3 levels in NDKD subjects compared to DKD and DKD+NDKD? Furthermore, the authors should suggest an explanation for the finding of the Cox regression that lower C3 predicts ESKD in DKD+NDKD group?
4. In the discussion section, alternating the discussions about the kidney outcome (lines 216-220, 227-228) with the ones about predictors of DKD or NDKD (220-222, 231-239) is hard to follow and misleading. They should be discussed separately.
5. The univariate Cox analysis results for the whole cohort are missing from table 3.
6. Please provide the Chi-square values for each of the Cox models and what entry method was used for the selection of the variables (backward?, forward?).

Validity of the findings

no comment

·

Basic reporting

Study rationale
The rationale for the study is not well explained. There is no reference to previous studies on this domain and how this study addresses the shortcomings of those inaccuracies in the present study in the introduction section.

Experimental design

This scientific study on Type 2 Diabetes Mellitus (T2DM) patients undergoing renal biopsy presents a comprehensive analysis of baseline characteristics, pathological features, and prognostic factors. While the study provides valuable insights, there are several aspects that warrant critical consideration:

Membranous Nephropathy:
My question is how the authors are attributing Membranous Nephropathy (MN) in NDKD. MN by its definition is an autoimmune disease attributed to thickening of the glomerular capillary walls due to immune complex deposition. Elevated levels of phospholipase A2 receptor (PLA2R) antigen is a diagnostic marker for MN, so the authors need to provide the method how it was concluded MN.

Sample Size and Exclusion Criteria:
The study initially involved 315 T2DM patients, but 47 were excluded due to non-compliance. This could introduce selection bias and potentially impact the generalizability of the findings. It would be beneficial to elaborate on the specific criteria for non-compliance.
Patient Demographics:
The study provides a breakdown of gender and age, but additional demographic factors such as socioeconomic status, lifestyle, and comorbidities could be relevant and may influence outcomes.
Follow-up Duration:
The average follow-up period is stated as 16.44 months. A longer follow-up might provide a more comprehensive understanding of the progression of renal complications in these patients.
Baseline Characteristics:
While the study analyses various baseline characteristics, it would be valuable to include information on factors like BMI, smoking status, and family history of diabetes, which can significantly impact the course of T2DM and its associated complications.
Pathological Classification:
The study identifies the most prevalent pathological types, but a more detailed discussion on the criteria for each classification and potential interobserver variability in histopathological assessment would enhance the rigor of the study.
Prognostic Factors:
The study identifies various prognostic factors for adverse renal outcomes in different groups. However, additional factors like medication adherence, lifestyle modifications, and access to healthcare resources may also be influential.

Comparative Analysis:
While the study compares its findings with previous research, it would be beneficial to discuss any discrepancies in methodology, patient populations, or diagnostic criteria that might contribute to differences in results.

Ethnic and Regional Considerations:
The study is based on a Chinese population, and it's important to acknowledge that findings may not be directly generalizable to other ethnic groups or regions with different prevalence rates and genetic predispositions.
Limitations and Future Directions:
The study briefly mentions its limitations, but a more detailed discussion of potential sources of bias or confounding, as well as avenues for future research, would add depth to the analysis.
Histopath sections:
Can image comparison of renal biopsy be furnished to improve confidence in the study and make it appear more appealing to the reader.

Validity of the findings

Novelty of experiment and inferences from data needs to be elaborated. And how this study is different from previous studies in this field needs to be explained at length, which is a major shortcoming of this manuscript.

Additional comments

Lack of histopathology sections is a major area of concern that should act as an impediment to the acceptance of the study. The data is clical data heavy and does not explore possible mechanisms in T2DM is another significant shortcoming of the study.

---

## Round 0.2 · Minor Revisions

Please try to rephrase more clearly the conclusions and please explain the values of C3.

Reviewer 1 ·

Basic reporting

All the raised issues were addressed. However, i identified other 2 minor issues that should be solved:
1. The phrase from lines lines 171-173 is not clear. I recommend to specify the groups for each result number. ”After analyzing the incidence of the endpoints, our study found that the number of patients with endpoints in the three groups were 13, 9, 12, with proportions of 17.57%, 8.26%, 14.12%”.
2. I recommend rephrasing a bit the conclusions, in particular this phrase ”Patients with DN were susceptible to getting the kidney endpoints.”

Experimental design

The authors addressed most of the issues, with one exception that I still think may be worth discussing - the significantly lower levels of C3 in DN subjects compared to NDKD and DN+NDKD, a finding that may seem strange. Maybe the following article will be helpful in this regard - https://link.springer.com/article/10.1007/s00592-017-1060-4#citeas

Validity of the findings

no comment

·

Basic reporting

The reporting has been convincing and detailed. Quality of English is professional, while I would request the authors may be to add more detail to the biopsy pictures attached in the supplementary files.

Experimental design

Experimental detailing is convincing with the updated incorporations and suggested changes which where made by the authors.

Validity of the findings

The findings are encouraging and may add value to the relevant research domain.

---

## Round 0.3 · accepted · Accept

There are no further comments.

Reviewer 1 ·

Basic reporting

No comment

Experimental design

No comment

Validity of the findings

No comment